# Innovative Preservation Methods Improving the Quality and Safety of Fish Products: Beneficial Effects and Limits

**DOI:** 10.3390/foods10112854

**Published:** 2021-11-18

**Authors:** Barbara Speranza, Angela Racioppo, Antonio Bevilacqua, Veronica Buzzo, Piera Marigliano, Ester Mocerino, Raffaella Scognamiglio, Maria Rosaria Corbo, Gennaro Scognamiglio, Milena Sinigaglia

**Affiliations:** 1Department of Agriculture Food, Natural Resources and Engineering (DAFNE), University of Foggia, Via Napoli 25, 71122 Foggia, Italy; barbara.speranza@unifg.it (B.S.); angela.racioppo@unifg.it (A.R.); antonio.bevilacqua@unifg.it (A.B.); mariarosaria.corbo@unifg.it (M.R.C.); 2UNCI AGROALIMENTARE, Via San Sotero 32, 00165 Roma, Italy; veronicabuzzo@gmail.com (V.B.); marigliano.piera@libero.it (P.M.); estermocerino3@gmail.com (E.M.); raffaella-rs@libero.it (R.S.)

**Keywords:** fish, spoilage, non-thermal atmospheric plasma, pulsed electric fields, pulsed light, ultrasound, electrolyzed water

## Abstract

Fish products are highly perishable, requiring proper processing to maintain their quality and safety during the entire storage. Different from traditional methods used to extend the shelf-life of these products (smoking, salting, marinating, icing, chilling, freezing, drying, boiling, steaming, etc.), in recent years, some alternative methods have been proposed as innovative processing technologies able to guarantee the extension of their shelf-life while minimally affecting their organoleptic properties. The present review aims to describe the primary mechanisms of some of these innovative methods applied to preserve quality and safety of fish products; namely, non-thermal atmospheric plasma (NTAP), pulsed electric fields (PEF), pulsed light (PL), ultrasounds (US) and electrolyzed water (EW) are analysed, focusing on the main results of the studies published over the last 10 years. The limits and the benefits of each method are addressed in order to provide a global overview about these promising emerging technologies and to facilitate their greater use at industrial level. In general, all the innovative methods analysed in this review have shown a good effectiveness to control microbial growth in fish products maintaining their organoleptic, nutritional and sensory characteristics. Most of the technologies have also shown the great advantage to have a lower energy consumption and shorter production times. In contrast, not all the methods are in the same development stage; thus, we suggest further investigations to develop one (or more) hurdle-like non-thermal method able to meet both food production requirements and the modern consumers’ demand.

## 1. Introduction

Fish were first vertebrates to appear on Earth, more than 500 million years ago and they can be considered the oldest, simplest and most abundant living vertebrates, in terms of number of species and populations. It is estimated that there are currently about 30,000 species, of which about 24,000 are known [1].

Although there are thousands of fish species, only a small part of these has economic value and supports fishing activities. Some species are used to produce protein supplements for human and animal consumption (*fishmeal*) or into the preparation of food products such as margarine, cosmetics, paints or even fertilizers, but most of them are appreciated for their meats and are a consistent part of the diet of the human being.

Due to their composition, fish products play a key role in a healthy diet for different reasons [2]. First, they are rich in proteins (up to 20% of weight) with high biological value due to the presence of essential amino acids. Then, the content in unsaturated fats (Omega-3), mainly eicosapentaenoic acid (EPA) and docosahexaenoic acid (DHA), allow fish products to maintain good health, reduce inflammation and control blood clotting and triglyceride levels. Finally, the presence of sodium and phosphorus supports the proper functioning of the thyroid gland (regulation of basal metabolism; consequently, it is important in the prevention of obesity) and calcium needed for bone formation, such as the main vitamins present (B and D) [2].

As for other foods, the achievement of high-quality standards is the main objective of the production chain of fish products and this purpose is inextricably linked to the freshness of the raw material. According to Oehlenschläger and Sörensen [3], freshness of fish means that a fish, in its entire characteristics, is not far away from those characteristics it had in the living state, or that only a short time has passed since the fish has been caught or harvested. Physical, chemical, microbiological and biochemical transformations happen immediately after death, resulting in a progressive loss of food properties in terms of taste and quality.

The high perishability of fish products is mainly due to their peculiar composition and structure, even if storage time and temperature are crucial factors for the final quality of the product. The major cause of fish perishability is attributable to the high content in non-protein nitrogen compounds and to the low acidity (pH > 6) of the flesh, which are conditions favourable to the growth of microorganism-producing metabolites that affect the organoleptic properties of the products. Nevertheless, the rate of spoilage is also due to the kind of fish species, the sanitary conditions on board and the amount of food in the guts [4].

Traditionally, the methods used to extend the shelf-life of fish products include fermentation, smoking, salting and marinating, or thermal treatments such as chilling, refrigeration, freezing, drying, boiling, steaming, etc. However, all these techniques are associated with undesirable changes, from a reduced nutritional value to worsened sensory attributes, which fight against the increasing demand of consumers for minimally processed foods with high quality. Thus, in recent years some alternative methods have been proposed as innovative processing technologies able to guarantee an extension of shelf-life while minimally affecting their organoleptic properties.

The present review aims to describe the primary mechanisms of some of these innovative methods applied to preserve quality and safety of fish products; namely, non-thermal atmospheric plasma (NTAP), pulsed electric fields (PEF), pulsed light (PL), ultrasounds (US) and electrolyzed water (EW) are analysed, focusing on the main results of the studies published over the last 10 years. After a description of the main mechanisms involved in fish spoilage, each innovative approach is addressed focusing on its limits and benefits, in order to provide a global overview about these promising emerging technologies and to facilitate their greater use at industrial level.

## 2. Fish Spoilage

In general, the rapid spoilage of fish after harvest is mainly due to different mechanisms, including (i) post mortem enzymic autolysis, (ii) microbial spoilage and (iii) oxidation of lipids.

Immediately after slaughter, the endogenous autolytic enzymes present in fish muscle become highly active and begin a proteolytic process which leads to protein decomposition and solubilization; peptides and free amino acids formed via autolysis, as well as biogenic amines formed through the action of decarboxylases, lead to fish spoilage [5].

Trimethylamine (TMA) is the main indicator of unpleasant ‘*fishy smell*’; it is a volatile nitrogenous base, produced post mortem in fish by the degradation of Trimethylamine Oxide (TMAO). TMA is below the detection limit in freshly caught fish, but some bacteria, such as *Shewanella putrefaciens*, *Aeromonas* spp., Enterobacteriaceae, *Photobacterium phosphoreum*, *Vibrio* spp., *Micrococcus*, *Acinetobacter*, *Moraxella,* use TMAO as an osmoregulant to avoid dehydration in marine environments and tissue waterlogging in fresh water by reducing TMAO to TMA, creating the ammonia-like off-flavours [4].

Other biological amines (BAs) are produced through microbial decarboxylation, including histamine, putrescine, cadaverine, spermidine and spermine [6]. Among BAs, histamine is the degradation product of histidine. Some histamine is produced by endogenous tissue enzymes in relatively small quantities, immediately after fish capture, while most histamine is produced by the bacterial flora [7]. The bacteria directly involved in the production of high levels of histamine are those possessing the enzyme histidine decarboxylase, such as *P. phosphoreum*, Enterobacteriaceae and Pseudomonadaceae (including *Morganella morganii* and *Klebsiella pneumoniae*, *Proteus* and *Pseudomonas* spp.) [8,9,10].

However, it is known that changes in the organoleptic characteristics of fish depend mainly on an increase in the microbial load. In particular, two groups of microorganisms can be distinguished in fish, namely, the indigenous or autochthonous microbiota and the exogenous or allochthonous microbiota. In fish from temperate or warm waters, the autochthonous microbial flora consists mainly of narrow aerobic or facultative aerobic mesophilic Gram-negative (*Pseudomonas* spp., *Moraxella*, *Acinetobacter*, *Flavobacterium*, *Xanthomonas* and *Vibrio*) and Gram-positive bacterial species (*Bacillus*, *Corynebacterium*, *Micrococcus* and *Lactobacillus*). Whereas, in cold-water fish, it consists of Gram-negative species in the surface mucus (mainly *Pseudomonas*, *Alteromonas*, *Photobacterium*, and *Shewanella*) and Gram-positive species in the intestinal contents (*Clostridium* spp.) [11,12,13]. The exogenous microbiota consists of typically terrestrial microbial species, such as *Enterococcus*, *Escherichia coli*, *Salmonella*, *Enterobacter*, *Klebsiella*, *Shigella* and *Yersinia*. This type of contamination mainly affects fish living near the coast contaminated by sewage from large urban agglomerations [14].

Finally, particularly in fatty fish, chemical oxidation of lipids is a common cause of spoilage which leads to the formation of all those compounds conferring the characteristic rancid off-flavours to spoiled fatty fish. Spoilage can be also caused by lipid hydrolysis through lipolysis. Lipolytic enzymes can either be endogenous of the fish itself (present in the fish skin, blood and tissue) or can be the product of the psychrotrophic microorganism’s metabolism. Regardless the origin of the enzymes, the fatty acids formed during hydrolysis interact with sarcoplasmic and myofibrillar fish proteins, causing denaturation and texture changes [15].

A summary of the main types of fish spoilage, together with the causes and the changes observed, is presented in Table 1. Even though the spoilage causes of fish can be explained through the above-mentioned processes, in general, all these mechanisms progress simultaneously, accelerating the overall spoilage of these products.

## 3. Innovative Preservation Methods Applied to Fish Products

The purpose of an optimal preservation method should be counteracting the causes of food deterioration maintaining its chemical (its composition), physical (its condition), organoleptic (taste, smell and colour) and nutritional (presence of proteins, fats and carbohydrates, vitamins, mineral salts and water) properties.

Therefore, besides traditional preservation methods, the great challenge of modern food technology is to develop less aggressive preservation processes, which keep the product ‘natural’, although with a lower shelf-life.

Non-thermal technologies are able to significantly inactivate microorganisms in food, extend shelf-life without significant changes in sensory perception and maintain the nutritional value of the processed food [16]. Among the main non-thermal inactivation techniques studied to be applied for fish products, non-thermal atmospheric plasma (NTAP), pulsed electric fields (PEF), pulsed light (PL), ultrasound (US) and electrolysed water (EW) are described in the following sections, the benefits and the limits of each method are highlighted and the potential positive or negative effect on the quality of the treated products is underlined.

### 3.1. Non-Thermal Atmospheric Plasma (NTAP)

In physics and chemistry, the term plasma is used to denote the state of an ionised gas. Plasma is considered the fourth state of matter, alongside the liquid, solid and gaseous states. While the presence of plasma on Earth is relatively rare (with the exception of lightning and the aurora borealis), in the Universe, it constitutes more than 99% of known matter; the upper layers of the Earth’s atmosphere (ionosphere), the outer gaseous layers of the Sun and stars and interstellar space are plasmas (natural plasmas).

A plasma can be generated artificially by supplying a gas with sufficiently high energy by means of lasers, shock waves, electric arcs, or electric and magnetic fields (glow discharge). There are two types of plasma, thermal and non-thermal atmospheric plasma (NTAP), depending on the conditions under which it is generated. Plasma generated at ambient pressure and temperature is called cold plasma (CP), atmospheric cold plasma (ACP), or non-thermal atmospheric plasma (NTAP) and it differs from thermal plasma obtained at higher powers and pressures. To generate NTAP, any type of energy (electrical, thermal, optical, radioactive and electromagnetic) can be used to ionise gases, but mainly electrical and electromagnetic fields are used.

Plasma has a neutral ionised gaseous form consisting of ions, free electrons, gas atoms and molecules, as well as UV photons depending on the process parameters and the gas used [17]. It is created after exposure of the gas to an electric field created between two electrodes (cathode and anode), separated by a small distance of 1 cm. The gases mainly used for plasma generation, which can also influence its properties, are oxygen (O_2_), nitrogen (N_2_), carbon dioxide (CO_2_) and noble gases, individually or in combination for optimum results. Oxygen seems to be more effective than the other gases due to its ability to cause greater oxidation of nucleic acids and amino acids [18].

Plasma technology has proven to be a successful tool in both the food sector (for the decontamination of abiotic food surfaces, such as packaging materials, foods in their final packaging and various food products) and medical sector [19,20]. In the last decade, NTAP technology has been widely used for the preservation of various food products, such as meat [21] and fresh agricultural products [22], while its use in fish and seafood is still limited [23,24,25].

The effectiveness of this technology obviously depends on many factors, such as voltage, frequency, treatment time, working gas composition (WGC), post-treatment/exposure time and sample surface area [24,26]. The type and concentration of reactive species (RESPE) produced, such as reactive nitrogen species (RNS) and reactive oxygen species (ROS), including ozone, peroxide, singlet oxygen and different types of nitrogen oxides (NxOy), are mainly responsible for the inactivation of microorganisms and depend on the above parameters.

However, this technology is only able to inactivate microorganisms on the surface of solid food, due to its poor penetration capacity. When a food has high microbial loads that form multiple layers of bacteria on the surface, the upper layers of cells protect those below and the decontamination effect is unfortunately not complete. Three mechanisms of action have been observed for the inactivation of microorganisms: (1) the direct disruption of the membrane or cell wall, with leakage of cellular components; (2) oxidative damage to membranes or intracellular components, such as proteins and carbohydrates; (3) damage to cellular DNA.

As shown in Table 2, NTAP technology has recently been proposed to inactivate many common pathogens in fish products (*Staphyloccocus aureus*, *Listeria monocytogenes*, *Salmonella* Typhimurium and Enteritidis, *Clostridium perfringens*, *E. coli*), various spoilage and spoilage microorganisms (*Pseudomonas*, hydrogen sulphide-producing microorganisms, Enterobacteriaceae), including yeasts and fungi (*Cladosporium cladosporioides* and *Penicillium citrinum*), standing up as a precious additional tool for the successfully decontamination of various food and seafood products.

A global overview of the analysed literature shows that NTAP could be suggested to extend the shelf-life of fish products due to its beneficial effects on the inactivation of microorganisms and enzymes (Table 2).

In general, a good effectiveness of this technology for microbial decontamination was always observed, also improving the food safety of treated products; the problem of the presence of *S. aureus*, for example, was found to be completely eliminated in semi dried mackerel pike by Puligundla et al. [27], who reported an inactivation of 3.08 log CFU/g by a treatment with plasma over 10 min. Similar results were recovered by Hajhoseini et al. [28] in fish nuggets and by Choi et al. [29,30] in dried squid shreds and black mouth angler.

Regardless the procedure used to generate plasma (corona discharge plasma jet, cold oxygen plasma, high voltage plasma, etc.), the inhibition of the main specific spoilage organisms (SSOs) of fish was generally recovered; in 2019, for instance, Albertos et al. [31] observed significant inhibitions of the total aerobic mesophiles and psychrotrophs, lactic acid bacteria, Enterobacteriaceae and *Pseudomonas* spp. by applying plasma at 80 kV for 5 min to herring (*Clupea harengus*). Similarly, plasma treatments reduced SSOs’ growth in mackerel fillets [32] and Asian sea bass slices [23,24,26,33].

Some researchers reported an additive effect by using different gas compositions (air, nitrogen, oxygen, argon, etc.) at different pressures for different time intervals, singly or in combination, for plasma generation and packaging types [34], or by pre-treating with chitooligosaccharides [35]; the NTAP effects were further enhanced in terms of microbial destruction and attributed to further diversification in the generation of reactive species.

Focusing on the reduction in enzymatic activity, chemically active species generated by NTAP could cause bond cleavage and side-chain modifications of enzymes, altering their secondary structure and, subsequently, their functionality [36]. Although few studies have analysed this issue in fish products, some authors have observed that 4 min treatments at 60 kV voltage decreased the proteolytic activity (50–64%) in large head hairtail (*Trichiurus lepturus*) [37] and in Argentine shortfin squid (*Argentinus ilex*) [38], thus slowing down the degradation of myofibrillar and collagenous proteins responsible of the fish softening and texture worsening. The inhibition of polyphenol oxidase by the application of NTAP in white shrimps was also recovered [39], as well as the inhibition of lipases, proteases and other enzymes [23,24,33,38,40,41]. However, this aspect needs to be further investigated.

The main limitation observed in almost all case studies analysed was lipid oxidation, which increases exponentially as exposure time and energy used raise; this can lead to the creation of short-chain fatty acids, aldehydes, acid hydroxides and ketoacids, thus causing off-flavours and off-odours during storage. However, some recent investigations found that pre-treatment of fish products with natural extracts rich in antioxidants could retard the rate of lipid oxidation in samples treated with NTAP [25,26,33,34,35].

In general, the sensorial qualities were unaffected by the application of cold plasma, even if samples treated for a longer time showed a lowered overall acceptability and a higher lipid oxidation.

As a final consideration, this technology has demonstrated an excellent ability to inactivate microorganisms without promoting their resistance or triggering deteriorative processes. Its application as a minimal processing method to preserve the quality of fish products is recommendable, since it offers very important advantages for food industries, namely, (1) it allows short processing times; (2) it is effective at low temperatures; (3) it is non-toxic; (4) its application reduces the consumption of water and chemical agents (less effluents).

Unfortunately, this method is not currently allowed to be used on foods as a great research effort is still necessary to accomplish its successful implementation at industrial level as a safe and effective alternative to traditional preservation methods. In fact, the difficulty in interpreting the data obtained by different studies using very diverse equipment and operating conditions, resulting in very different plasmas in terms of properties and, consequently, with very different antimicrobial effectiveness, hinders its application in the food industry.

**Table 2 foods-10-02854-t002:** Application of non-thermal atmospheric plasma (NTAP) technology to fish products; main results and limitations of the technology considered.

Fish Product	Treatment Conditions	Tested Microorganisms	Results	Limit	Reference
Dried filefish fillets (*Stephanolepis cirrhifer*)	Cold oxygen plasma (COP); treatment time, 3–20 min.	*Cladosporium cladosporioides* *Penicillium citrinum*	Reduction >1 log_10_ CFU/g was observed in the fillets treated with COP for >10 min.	Exposure to 20 min of treatment showed an increase in lipid peroxidation and a decrease in overall sensory acceptance.	[18]
Dried squid shreds	The corona discharge plasma jet (CDPJ) was generated using 220 V AC power with an output voltage of 20 kV DC, at a current of 1.50 A and a frequency of 58 kHz.	Total aerobic count Marine bacteria *Staphylococcus aureus*	Aerobic bacteria, marine bacteria and *St. aureus* were inactivated by 2.0, 1.6 and 0.9 log units, respectively. Additionally, a 0.9 log reduction in yeasts and mould contaminants was observed.	A change in moisture content and thiobarbituric acid concentration was observed. All other physico-chemical and sensory properties tested were unaffected.	[29]
Fresh mackerel fillets (*Scomber scombrus*)	Plasma was generated using voltages of 70 and 80 kV for different treatment times (1, 3 and 5 min).	Total aerobic count Psychrotrophic bacteria *Pseudomonas* spp. Lactic acid bacteria	There was no significant (*p* > 0.05) reduction in the total aerobic mesophilic count, whereas psychrotrophic bacteria, LAB and Pseudomonas counts were significantly (*p* < 0.05) reduced due to DBD.	Changes in immobilised and extra-myofibrillar water were observed. Mackerel was more susceptible to lipid oxidation. There was no negative influence on physico-chemical parameters such as pH and colour.	[42]
Chub mackerel (*Scomber japonicus*)	Plasma was generated using a voltage level of 0, 10, 20, 30, 40, 50, 60 and 70 kV and treatment times of 0, 15, 30, 45, 60 and 75 s.	Endogenous microbiota	Under optimal conditions at 60 kV for 60 s, the microbial count decreased substantially with a slowdown in bacterial proliferation and a reduction in the production of volatile bases and oxidation compounds. There was also a delay in myofibrillar protein degradation and an improvement in microstructure stability. The shelf-life was extended to 14 days against 6 days recovered for samples without this treatment.	Slight alteration of the chemical composition.	[43]
Smoked salmon	UV-C at 254 nm and a high-voltage plasma jet at 1 kHz were used, at predetermined time intervals (0, 1, 2 and 4 min), with intensities of up to 500 mJ/cm^2^.	*Listeria monocytogenes* *Listeria innocua* *Salmonella* Typhimurium *Salmonella* Enteritidis *St. aureus* *Escherichia coli* O157:H7 *Aeromonas hydrophila* *Plesiomonas shigelloides*	An additive lethal effect of the two techniques was found, with a reduction of 0.5–1.3 log CFU/g in the microbial population	High-energy treatments and long exposure times have caused significant changes in the appearance and oxidation of lipids	[44]
Asian sea bass slices	Plasma was generated using a voltage of 80 kV for 0, 2, 5, 7.5 and 10 min at room temperature (28 ± 2 °C).	Total viable count Psychrophilic bacteria H_2_S-producing bacteria Enterobacteriaceae *Pseudomonas* *Clostridium perfringens* Lactic acid bacteria	In treated samples, total viable count (TVC) was lower than the acceptable limit (log 10^6^ CFU/g sample) within 18 days. The growth of various pathogenic and spoilage bacteria, including psychrophilic bacteria, *Cl. perfringens* (not detected), lactic acid bacteria (3.77–4.37 log CFU/g), Enterobacteriaceae (4.03–4.50 log CFU/g), Pseudomonas (6.62–6.82 log CFU/g) and hydrogen sulphide (H_2_S)-producing (4.04–5.05 log CFU/g) bacteria, of treated slices was lower than control samples. A 5-min treatment extended shelf-life to 12 days against 6 days recovered for samples without this treatment.	Pronounced lipid oxidation was observed in the 7.5 and 10 min treatments. There was also a reduction in the amount of PUFA and MUFA fatty acids by 28–64% and 40–46%, immediately after treatment and after 12 days of storage.	[23,24,33]
Refrigerated Asian Sea bass slices	Plasma was generated with an input voltage of 230 V at 50 Hz and an output voltage controlled within 0–120 kV.	Total viable count Psychrophilic bacteria Lactic acid bacteria *Pseudomonas* spp. H_2_S-producing bacteria Enterobacteriaceae *Cl. perfringens*	The shelf-life was extended to 15 days, while the control (kept in air) had shelf-life of 6 days.	Pronounced oxidation of proteins and lipids.	[26]
Asian sea bass slices (*Lates calcarifer*)	Cold atmospheric plasma was generated with a mixture of argon and oxygen (90% Ar/10% O2) for 5 min and used in combination with chito-oligosaccharides (COS) at different concentrations (0.05, 0.1 and 0.2 g/100 g).	Total Viable Count Psychrophilic bacteria Enterobacteriaceae *Pseudomonas* spp. H_2_S-producing bacteria Lactic acid bacteria *Cl. perfringens*	Reduction in *L. monocytogenes*, between 1.21 and 1.52 log CFU/g; reduction in *S.* Typhimurium, between 1.44 and 1.75 log CFU/g. The thiobarbituric acid reactive substances (TBARS) and peroxide values (PV) of treated samples were reduced. Sensory acceptability was improved.	No negative effects were found.	[34]
Grass carp (*Ctenopharyngodon Idella*)	Plasma was generated using air as the feed gas, at a current and frequency of 1.05 A and 10 kHz, respectively, under atmospheric pressure and an ambient temperature of 25 °C. The applied voltage was 70 V with a peak input power of 73.5 W.	*L. monocytogenes* *S.* Typhimurium	Logarithmic reductions were observed between 1.21 and 1.52 for *L. monocytogenes* and between 1.44 and 1.75 for *S.* Typhimurium.	Reduction in pH and increase in total acidity level in samples and change in colour.	[45]

### 3.2. Pulsed Electric Fields (PEF)

PEF is an emerging technology that involves the delivery of short high-power electrical pulses (microsecond) to a product placed in a treatment chamber, confined between electrodes. The process produces modest thermal increases without causing any effect in the product. The application of an external electric field to biological cells (animal, plant or microbial) causes damage to the cell membrane. To date, a number of theoretical models have been suggested, but there is still no clear evidence of the mechanism of action at the cellular level. The most accepted theory is the electromechanical model introduced by Zimmermann et al. [46], which considers the cell membrane to be a capacitor with a low dielectric constant. Free charges of opposite polarity are present on both sides of a membrane (inner and outer), resulting in a natural transmembrane potential. Exposure to an electric field induces accumulation of charges inside and outside the cell across the membrane and thus an increase in transmembrane potential.

When the transmembrane potential exceeds a critical value, there is a rapid electrical collapse of the cell membrane, whose structure changes, with an increase in permeability, loss of cellular components and collapse of the proton motive force. Charges with opposite signs are formed on both sides of the membrane, compressing it and forming pores. The breaking of the membrane can be reversible or irreversible, depending on the intensity of the treatment [47]. When the induced pores are small compared to the area of the membrane and are generated with a low intensity PEF treatment, the effect is reversible [48]. Based on this phenomenon, electroporation (permeabilization of the cell membrane caused by an external electric field) has been studied in practical applications on various biological systems in the fields of medicine, biology and food processing.

Besides being a non-thermal alternative, this technique proved to have a good impact on the microstructure of muscle foods [49,50], without affecting physical, organoleptic and functional characteristics [51,52]. For example, Gudmundsson and Hafsteinsson [53] observed that salmon fillets improved significantly their texture and microstructure when subjected to a mild PEF treatment (<2 kV/cm, 20–40 pulses). Contrarily, no positive effects were observed on the tenderness of some types of mussels and molluscs [54].

Although the technique was known 50 years ago, PEF can be still considered an emerging technology, because its industrial applications are recent. Initially, the use of PEF in food processing has attracted great interest especially as a method to improve the extraction of specific components or improve drying efficiency [55,56]. The technology is, in fact, well suited for the extraction of high-value compounds from fish by-products, as it destroys only the biological cells in the food matrix with a high extraction efficiency, also compared to other methods [57]. Surprising results were recovered by using PEF to extract proteins from mussels [58], calcium and chondroitin sulphate from fishbones [57,59] and proteins from abalone viscera [60].

However, more recently, the use of PEF has been suggested also as a novel preservation method, due to its capacity to rapidly inactivate microorganisms, producing foods with great nutritional and sensory quality [61,62,63]. It has been also shown that using PEF in combination with other non-thermal technologies such as UV irradiation, microwaves, high-intensity light pulses (HILP) and high hydrostatic pressure (HHP) increased microbial inactivation [64,65].

From the few papers available in the literature, PEF appear to have no negative effects on sensory and nutritional qualities when applied to fish products, although this aspect merits further investigation. In contrast to NTAP (which is mainly used on solid foods), PEF technology is preferred for treating liquid foods; this explains the limited number of studies on fish and seafood. In 2020, Shiekh et al. [25] tested the effect of pulsed electric fields on microbiological changes in Pacific white shrimp (*Litopenaeus vannamei*). The samples were stored at 4 °C for 10 days and were treated every second day with PEF at different densities. The treatment conditions were PEF-T1 (5 kV/cm, 200 pulses), PEF-T2 (10 kV/cm, 400 pulses) and PEF-T3 (15 kV/cm, 600 pulses) with the PEF specific energy of 54,214 and 483 kJ/kg, respectively. The most effective treatment for bacterial inactivation was the high-intensity treatment (PEF-T3), which resulted in a low microbial load until the end of storage (approximately 4.58 log CFU/g). The technology appeared as an effective method to inhibit psychrophilic bacteria, which are the main cause of shrimp spoilage during cold storage. In addition to the microbiological result, chemical inhibition of the enzyme polyphenol oxidase was observed, resulting in improved sensory and nutritional properties.

The limitation to its wider use in the food industry lies in the high initial cost of the equipment, as well as the fact that studies on its application to food products have mainly been conducted on liquids with low electrical conductivity. Although PEF is a non-thermal treatment, when used at high intensity, there is a significant increase in temperature, which must be considered with sensitive compounds such as proteins [61,66]. Further investigations should be performed to evaluate the impact of this technology on the quality parameters of fish products (e.g., tenderness, colour, oxidation, weight loss and water retention capacity). Moreover, the inefficiency of this technique against the reduction in natural enzymes present in fish is another shortcoming of this emerging technology [67].

Nevertheless, the transfer of PEF technology to the fish industry would be highly favourable due to the low energy consumption and short processing times required.

### 3.3. Pulsed Light (PL)

Pulsed light (PL) is a non-thermal technology, approved by the FDA (Food and Drug Administration), which involves the emission of short flashes of light in a broad spectrum [68]. PL technology was first used in the medical field to sterilise medical devices and then in water purification processes; recently, it has also found new applications in air sanitation. In 1996, the FDA approved the use of PL technology for food production, processing and handling processes [69]. It is recommended to use the xenon lamp with surface emission of wavelengths between 200 and 1100 nm, with a cumulative treatment not exceeding 12 J/cm^2^ and a pulse width not exceeding 2 ms. In the food industry, pulsed light technology is mainly used for ready-to-eat products, meat and fish products, or dairy products, which are subject to rapid spoilage and require delicate preservation measures. The decontamination effect of PL treatments is mainly due to the photochemical changes caused by UV-C radiation on microbial DNA, in combination with the photothermal and photophysical damage caused to cells by water vaporisation and membrane destruction [70]. The effectiveness of this method has been recognised against Gram-positive and -negative bacteria, as well as fungal spores, and the lethal effect is greater than UV treatment applied in continuous. In particular, it has been shown that Gram-positive bacteria are more resistant than Gram-negative bacteria and fungal spores show higher resistance than bacteria [71], although Gómez-López et al. [72] reported opposite results. However, each microorganism has a different sensitivity to treatment and this may be related to differences in the composition of the bacterial cell wall and their protective and repair mechanisms against damage [71]. The potential of pulsed light treatment depends on many factors, such as the exposure time, variations in the power of the UV source (which affects the electromagnetic wavelength), the presence of particles that can protect microorganisms from UV and the ability of the microorganisms to resist the radiation during exposure. High power, long treatment time and the closer distance between target and flash lamp cause an increase in microbial reduction but a consequent loss of quality, so it is necessary to find the optimal treatment conditions to improve microbiological safety without affecting food quality [73,74]. For example, in a study performed on salmon fillets, the application of PL at 5.6 J/cm^2^ at intervals of 3 and 5 cm for 60 s, caused an increase in the product’s temperature up to 100 °C, resulting in good microbial reduction but significant changes in colour and quality. However, the same treating time (60 s) applied at a longer distance (8 cm), was able to reduce *E. coli* O157:H7 and *L. monocytogenes* by approximately 1 log CFU/g, without compromising the quality of the fish product [75].

According to the study by Mandal et al. [70], PL technology is an effective, fast and mild decontamination method and its applications are increasing not only for food contact surfaces, but also for the decontamination of packaging materials. However, this technology, due to the non-uniform shape and opacity of the products, cannot be used for sterilisation processes, but only for reducing the microbial load. In addition, the application of PL is particularly difficult in the case of foods in granular form, such as cereals or spices, due to the shadow effect of the surfaces, which does not allow light to reach the microorganisms.

As reported in Table 3, PL has been recently suggested to be used in fish products for the inactivation of altering bacteria (*P. phosphoreum*, *Serratia liquefaciens*, *S. putrefaciens*, *Brochothrix thermosphacta*, *Pseudomonas* group I and *Pseudomonas* groups III and IV) and *L. monocytogenes*, which has proven to be more resistant. A careful review of the literature also highlighted a possible use of PL technology for the decontamination of packaged fish products, e.g., in vacuum packs, where the product surface is completely or almost completely exposed to the pulsed light. Unfortunately, the use of this technique resulted in a short-term decline in sensory quality. To maintain the attributes of the product, lower fluences should be applied, but this would cause lower inactivation of pathogens. Moreover, PL used alone could (1) induce surface discoloration, (2) accelerate product senescence and oxidative processes, (3) increase lipid oxidation as hydrogen peroxide and superoxide radicals are formed indirectly by UV light and (4) have an impact on colour (since the peroxide formed during prolonged treatment can diminish pigments) and on protein fragmentation with an impact on texture.

Therefore, further research is needed to overcome these limitations and to analyse other aspects, such as the ability of PL to increase the nutritional attributes of products, leading to new opportunities for its use as a biofortification technology.

### 3.4. Ultrasound (US)

Ultrasound is one of the innovative non-thermal techniques that is proving to be very successful in the food sector where it is actually used for freezing, cutting, drying, homogenisation, degassing, foaming, filtration and extraction processes. More recently, it has been also proposed as an alternative to heat treatments to control microbial growth [76,77,78,79]. Ultrasonic waves used in the food industry are low energy, high frequency (16–100 kHz) waves. Any type of system used for US production consists of three parts: (1) a current generator that supplies electricity at the desired frequency to the transducer; (2) a transducer or converter, which converts electrical energy into mechanical vibrations (pressure waves) that are conveyed into a probe; (3) a probe that amplifies the vibration produced forming the sonication site that can be continuous or discontinuous. Typical ultrasonic systems are the ultrasonic bath, ultrasonic probes, parallel vibrating plates and radial vibrating systems. The mechanism behind sonication is the well-known phenomenon of cavitation, i.e., the repeated creation of microbubbles inside a liquid, followed by their implosion. The pressure resulting from these implosions causes the main bactericidal effect of ultrasound, which consists of a thinning of cell membranes, localised heating and production of free radicals [76,77,78,79]. The effectiveness of the treatment depends on several factors, such as type of microorganism treated, amplitude of the ultrasonic waves, exposure/contact time, volume and composition of the food to be treated and temperature of the treatment. The literature reports that Gram-positive cells are more resistant to ultrasound than Gram-negative cells and this may be due to the structure of the cell wall. In addition, vegetative cells are more susceptible to bacterial spores. To make the action of ultrasound on microorganisms more effective, sonication is often combined with other treatments. It is common to use mild heat treatments (thermo-sonics), high pressures (mano-sonics), or both (mano–thermo-sonics) [80].

Ojiha et al. [81] proposed a unifying mechanism to address the effect of US on cells; this mechanism is known as sonoporation and relies upon six different ways of acting on cells: push, pull, acoustic streaming, jetting, translation and cavitation; the combination of these mechanisms causes the disturbance of microbial homeostasis, morphological changes and the disruption of both cell wall and cell membrane.

US has been used for microbial inactivation in a wide variety of media, but only a few studies have focused on fish products (Table 3). In general, the ultrasonic waves used in fish studies are high-energy, low-frequency waves of 20–100 kHz. In a recent investigation conducted by Mikš-Krajnik et al. [82], US was used to decontaminate salmon fillets (*L. monocytogenes*, total bacterial count, yeast and moulds). The results showed different effects for each microorganism; *L. monocytogenes* and coliforms were reduced by 0.4 and 0.3 log CFU/g, respectively, while there was no significant reduction in total bacterial count, yeasts and moulds. The presence of irregularities on the surface of the fish fillet was speculated as the cause of these differences. With regard to quality indicators, an increased moisture content and a slight colour change were observed.

When applied as a single-use technology, this technology cannot achieve the 5 log reduction in compliance with the FDA (2004) requirements [69]; thus, its use is actually suggested to be combined with mild thermal treatments. The treatment appears better than traditional pasteurisation techniques, due to the absence of negative effects on the nutrient content and physical characteristics of treated food products. However, the effect of the application time should be considered; longer exposure times (more than 60 min) are not recommended. Contrarily, when this technology is applied for short times (about 20 min), it improves the texture and does not affect the physico-chemical characteristics [83].

Similar to PEF, US is proposed to be industrially used as a method of extracting compounds from plant and animal tissues; for instance, the cavitation effect generated by US has been exploited to extract lipids and carotenoids from crustacean processing by-products [84,85]. More, the use of US is suggested in the fish industry to decontaminate knives used in cutting operations. This can be seen as a comprehensive approach to improve the quality of fish products [86].

Evaluating the results of recent studies, US can be seen as an interesting technology to improve the stability of fish products. The combination with other technologies, such as PL or EW, could be an added value to further studies and to create new opportunities to implement this technology at industrial level, also considering that the method is fast, reliable, relatively cheap and easy to use.

### 3.5. Electrolysed Water (EW)

Among the relatively new proposals, electrolytic water (EW) is attracting interest as a non-thermal technique in the food industry and agriculture. Similar to all the innovative methods mentioned above, EW is safer and more effective than traditional chemical agents, to which microorganisms are becoming increasingly resistant. In fact, it is considered as a new non-thermal and environmentally friendly sanitiser.

EW was initially developed in Japan [87] and has been recognised as having a strong bactericidal effect on the main food pathogens; it is generated by a process of controlled diaphragm electrolysis produced by passing a salt solution through an electrolytic cell, where the anode and cathode are separated by a membrane. During the electrolytic process, NaCl splits into metallic sodium (Na) and chlorine gas (Cl_2_), while water (H_2_O) splits by electrolysis into hydrogen (H_2_) and oxygen (O_2_). The negatively charged ions Cl- and OH- lose their electrons through the generator anode, while, during this oxidation, hypochlorous acid (HClO), hypochlorite ion (ClO^−^), hydrochloric acid (HCl), gaseous oxygen (O_2_) and gaseous chlorine (Cl_2_) are generated. Conversely, positively charged ions (Na^+^ and H^+^) gain electrons pushed out of the cathode, where reduction occurs, resulting in the generation of sodium hydroxide (NaOH) and hydrogen gas (H_2_) [88]. Within the chamber, two types of EW are produced, namely, at the anode, acidic electrolysed water (AEW) or electrolytic oxidising water (EOW), with a pH value of 2–3, oxidation–reduction potential (ORP) >1100 mV and chlorine concentration of 10 to 90 ppm, while, at the cathode, basic electrolytic oxidising water (BEW), with a pH value between 10 and 13 and an oxidation–reduction potential of 800–900 mV. Another type of EW is neutral electrolysed water (NEW), with a pH value of 7–8 and an ORP of 750–900 mV. The effectiveness of the EW generated varies depending on the type and concentration of the solution, the voltage and current value, the water flow and the electrolysis time.

The antimicrobial activity of EW has been widely demonstrated against various food-borne microorganisms, such as *Pseudomonas aeruginosa* [89,90], *S. aureus* [90], *E. coli* O157: H7 [91], *S.* Typhimurium [92], *L. monocytogenes* [91,93], *C. jejuni* [93] and *V. parahaemolyticus* [94]. It is also effective against spores, fungi and viruses present in food, environment and food processing plants. The antimicrobial activity and mechanism of action of EW against bacteria are not yet fully described. However, it is known that chlorine and reactive oxygen can break down the microbial cell membrane and cause oxidative DNA damage.

EW has various applications in the food industry; one of the most significant is in the seafood industry, although limited effectiveness in decreasing the bacterial load on seafood at room temperature has been demonstrated. The treatment of fish products with various types of EW highlights great results for microbiological quality, but also good results in inhibiting pH changes, formation of total volatile basic nitrogen (TVB-N) and activity of the enzyme polyphenol oxidase (PPO) (see Table 4). Significant reductions in *L. monocytogenes* were observed both in salmon fillets and in carp skin after a treatment with AEW for 15 min [95]. Similarly, the effect of AEW was found to be very effective for reducing *V. parahaemolyticus* and *E. coli* O157:H7 on tilapia skin [96]. Although the application of EW on fish products appears a promising technique for reducing the total count of pathogenic and spoilage bacteria, at the same time, it has shown some undesirable effects on the organoleptic quality and nutritional value of food [97,98,99,100]. Since food safety must be accompanied by sensory quality, to overcome these limitations, a combination of two or more preservative and sanitizing technologies in low quantities is suggested. Consequently, several studies in the literature report the combined use of EW with MAP packaging, chitosan, or natural antimicrobial solutions [101,102,103,104]. The combined treatments compared with individual treatments showed a better preservative or even synergistic bactericidal effect, thus suggesting that the food industry would greatly benefit by adopting treatment procedures involving combinations of EW and other treatments.

However, limitations such as corrosion to equipment and detrimental effects on the quality of the treated food products, environment and human health have to be considered [105]. In addition, the high concentration of NaCl used for the production of acidic electrolytic water may lead to an increase in salinity in the pre-treated seafood. This can be perceived by consumers, thus lowering sensory acceptability. Finally, the chlorine ion can interact with other main components of food, thus influencing the texture of food and inducing certain reactions that occur during processing [106].

EW is successfully used in Russia and Japan as substitute of chemicals, whereas it is slowly obtaining acceptance in the US and other countries [106]. In the European Union (EU), EW can only be applied to “drinking water” and its use on food products such as fish is not yet permitted. In the near future, it is desirable that most of the industry start using EW, since these solutions are relatively simple in composition and not toxic [106]. Through further research, an advanced and dynamic EW production system able to reduce all the current limitations could be developed, also including processing settings providing the application in HACCP and sanitation SOP systems.

**Table 3 foods-10-02854-t003:** Application of pulsed light (PL) and ultrasound (US) technology to fish products; main results and limitations of the technologies considered.

Pulsed Light (PL)					
Fish Product	Treatment Conditions	Tested Microorganisms	Results	Limit	Reference
Beef and tuna carpaccio	A pulsed light device equipped with two xenon lamps was used. The lamps emitted flashes of 150 J, equivalent to a fluence of 0.175 J/cm^2^ per pulse. The pulse period was 250 μs and also the spectral output of the lamp corresponded to 30% UV light (12% UV-C, 10% UV-B and 8% UV-A), 30% infrared radiation and 40% visible light.	*Listeria monocytogenes* *Escherichia coli* *Salmonella* Typhimurium *Vibrio parahaemolyticus*	The application of pulsed light at the highest fluences tested (8.4 and 11.9 J/cm^2^) improved the microbiological safety of the product. Reductions from 2 to 6 log cfu/cm^2^ were achieved.	The application of pulsed light at the highest fluences tested compromised the sensory quality in the short term. To maintain product attributes, lower fluences should be applied, albeit at the expense of less inactivation of the tested pathogens.	[107]
Refrigerated tilapia (*Oreochromis niloticus*) fillets	After active packing with O_2_, the fillets were subjected to UV-C radiation in an apparatus containing six 30 W and six 55 W lamps. Exposure times were measured every 5 s up to doses of 0.102 ± 0.001 J/cm^2^ and 0.301 ± 0.001 J/cm^2^.	Enterobacteriaceae Total aerobic count	The O_2_ scavenger, UV-C doses (0.102 and 0.301 J/cm^2^) and combinations of these preservation methods retarded bacterial growth and the formation of TVB-N and ammonia, increasing the shelf-life of chilled tilapia fillets to 14–16 days against 9 days recovered in untreated samples.	UV-C used alone induced negative changes in colour, texture and oxidative processes. The O_2_ scavenger has proven to be an effective and simple alternative to reduce the negative effects of UV radiation.	[108]
**Ultrasound (US)**					
**Fish Product**	**Treatment Conditions**	**Tested Microorganisms**	**Results**	**Limit**	**Reference**
Refrigerated carp (*Ctenopharyngodon idellus)* fillets	Carp fillets were treated with chito-oligosaccharides (1%) and treated with ultrasound at 40 kHz for 10 min.	*Aeromonas* *Shewanella*	In comparison with control, treatments had positive effect on reducing the accumulation of TVB-N, off-taste nucleotides and biogenic amines, inhibiting microbial growth and maintaining sensory quality of fillets. The shelf-life of treated fillets was extended by nearly 2 days when compared to untreated samples.	No negative effects were observed.	[109]
Thawed cod fillets	The treatment was carried out in an ultrasound bath using three different powers, 29.4 W/kg (100%), 14.7 W/kg (50%) and 2.9 W/kg (10%), for 20 min.	Total aerobic count Mesophilic bacteria Seafood spoilage organisms (SSOs) Enterobacteriaceae Proteolytic bacteria	The US-assisted hydration process was able to control microbial growth without compromising the sensory quality properties of the cod fillets.	No negative effects were observed.	[110]
Salmon (*S. salar*), mackerel (*S. scombrus*), cod (*G. morhua*) and hake (*M. merluccius*)	A low-frequency (30 kHz) ultrasonic bath and a transferred ultrasonic power of 51.41 W/l at 14 °C was used.	Total mesophil and psychrophil counts *Pseudomonas* spp., Enterobacteriaceae	US treatment was able to significantly reduce microbiological counts in oily fish species with reductions of up to 1.5 and 1.1 log CFU/g for psychrophilic and mesophilic viable counts observed in salmon and mackerel, respectively. Lower reductions were observed in white fish species. Lipid content did not change, whereas significant reductions in TBARS values were observed in salmon. Moisture levels increased by 8%.	Colour changes in salmon samples were observed.	[111]

**Table 4 foods-10-02854-t004:** Application of electrolyzed water (EW) technology to fish products; main results and limitations of the technology considered.

Fish Product	Treatment Conditions	Tested Microorganisms	Results	Limit	Reference
Salmon fillets	The anolyte contained approximately 300 mg/L of free chlorine, an oxidation-reduction potential of 850 mV, a neutral pH (7.0 ± 0.1) and a residual chloride level of <0.5%. The solution was diluted to 50% and 15% (v/v).	Total aerobic counts Coliform *Pseudomonas* spp.	The use of the 15% or 50% solution for treatment significantly reduced the initial microbiota (approx. 1–2 log colony-forming units) during storage and significantly extended the shelf-life of the fillets by 2 and 4 days, without affecting the overall quality of the fillets, both raw and cooked.	No negative effects were found, instead, the significant increase in shelf-life and quality of fillets was corroborated by raw and cooked sensory evaluation.	[112]
Live clam (*Venerupis philippinarum*), mussel (*Mytilus edulis*)	Two types of acidic electrolyzed water (AEW) were used for treatment time, strong (SAEW), with an available chlorine concentration of 20 mg/L, pH 3.1 and an oxidation–reduction potential of 1150 mV, and weak (WAEW), with 10 mg/L of chlorine, pH 3.55 and potential of 950 mV.	*Escherichia coli* O104:H4 *Listeria monocytogenes* *Aeromonas hydrophila* *Vibrio parahaemolyticus* *Campylobacter jejuni*	SAEW and WAEW showed significant inhibitory activity against inoculated bacteria in each shellfish species. SAEW showed the highest antimicrobial activity, with reductions from 1.4 to 2.2 logarithmic cycles for the different microorganisms.	Weak electrolysed water showed fewer effective results than strong electrolysed water.	[113]
Atlantic Salmon (*Salmo salar*)	Acidic electrolysed water with pH 2.7, oxidation–reduction potential 1150 mV and free chlorine concentration of 60 ppm (generated at 9–12 V direct current for 15 min). Neutral electrolyzed water (NEW) with active hypochlorous acid (275 ppm) was electrochemically generated and diluted to obtain a solution with an available free chlorine content of 60 ppm, a pH of 6.8 and a potential of 786 mV.	*L. monocytogenes*	AEW and NEW showed strong antimicrobial properties against *L. monocytogenes.* The initial inoculation was 7.9 log CFU/g, which was reduced to 2.3 log CFU/g in samples treated with NEW at 65 °C for 10 min. By increasing the temperature and exposure time, the efficacy of electrolysed water increased significantly.	Further studies are needed on the effect of NEW and mild heat treatment on lipid oxidation, changes in amino acids, nutritional value and product preservation.	[114]
Shrimps	The samples were inoculated with *Vibrio parahaemolyticus* and subsequently treated using AEW1, with 51 mg/L chlorine, AEW2, with 78 mg/L chlorine, or organic acids (2% AA and 2% LA), for 1 min or 5 min under different treatment conditions.	*V. parahaemolyticus*	AEW treatment at 50 °C revealed a 3.1 log CFU/g reduction in *V. parahaemolyticus.*	The treatment significantly influenced the physico-chemical properties (pH, ORP, ACC).	[106]
Shrimp (*Litopenaeus vannamei*)	AEW was obtained by electrolysis of a 0.1% sodium chloride solution using a strongly acidic electrolyte water generator. AEW was frozen for 24 h	Total viable count	AEW ice was able to inactivate the bacterial load on raw shrimp; the total viable bacterial populations were reduced by 1.5 log CFU/g after 24 h. AEW ice also inhibited TVBN formation and PPO activity.	No negative effects were found.	[115]
American shad (*Alosa sapidissima*)	The electrolyzed oxidizing water (EOW) was generated with NaCl (0.1%) and deionised water, with a pH of 2.4, a potential of 1185 mV and a free chlorine level of 70 and 80 ppm. Dietary chitosan was used as a 2% (w/v) coating solution.	Total viable count Total aerobic count	The results of microbiological, physico-chemical (pH, TVBN, TBA, texture and colour) and sensory analyses revealed that the combined treatment successfully inhibited microbial growth, protein degradation and lipid oxidation and did not change texture, colour, or sensory characteristics during storage. This treatment extended the shelf-life of American shad fillets by 9–10 days during refrigerated storage.	No negative effects were found.	[101]
Pacific white shrimp (*Litopenaeus vannamei*)	In this study, weakly acidic electrolyzed water (WAEW) was used in combination with the modified atmosphere packaging (MAP). The WAEW had a pH of 6.4 and 6.6, an oxidation–reduction potential between 520 and 540 mV and an available chlorine concentration of 6.4 and 6.5 mg/L.	Total aerobic count *Staphylococcus aureus*	WAEW and MAP (40% CO_2_, 0% O_2_, 50% N_2_; 30% CO_2_, 20% O_2_, 50% N_2_) exerted a significant effect on spoilage inhibition, controlling microbial growth, increase in TVBN, TMA and TBARS and degradation of sensory properties.	No negative effects were found.	[116]
Raw trout	Acid electrolyte oxidising water (pH 2.30 and free chlorine 38 ppm), sterile distilled water was tested for 0 (control), 1, 3, 5 and 10 min at 22 °C.	*E. coli* O157:H7 *Salmonella* Typhimurium *L. monocytogenes*	The use of AC-EW was found to be the most effective treatment in reducing *E. coli* O157:H7, *S.* Typhimurium and *L. monocytogenes*. The level of reduction ranged between ca. 1.5 and 1.6 logs for *E. coli* O157:H7 and *S.* Typhimurium, and 1.1–1.3 logs reduction for *L. monocytogenes*.	There was no complete elimination of inoculated pathogens after treatment.	[117]
Cold smoked atlantic salmon (*Salmon salar*)	Electrolysed water (pH 2.7; ORP 1150 mV; free chlorine 60 ppm) was generated at 9 and 12 V DC for 15 min. Samples inoculated with the target bacteria, were treated with EW at different temperatures (20, 30 and 40 °C) and at different times (2, 6 and 10 min).	*L. monocytogenes*	Treatment prior to cold smoking at 40 °C for 10 min was able to reduce the cellular load of *L. monocytogenes* by 2.85 log CFU/g without causing any significant change in sensory properties.	No negative effects were observed.	[104]
Squid	The slightly acidic electrolyzed water (SAEW) was prepared by electrolysis of an aqueous mixture containing 0.2% NaCl and 0.04% HCl, then frozen immediately.	Endogenous microbiota	SAEW ice has been shown to be able to inhibit bacterial reproduction during storage by 1.46 log CFU/g, extend shelf-life and maintain good squid quality for the entire observation period (6 days).	No negative effects were found.	[118]
Raw salmon fillets	Electrolytic acidified water (AEW) was obtained at a constant current of 10 A by electrolysis of a sodium chloride solution (0.1%, w/v), pH (2.6 ± 0.2) and oxidation/reduction potential of 0.1140 ± 30 mV. The fillets were treated for 1, 5, or 10 min at room temperature.	*L. monocytogenes* Natural microbiota	The treatment reduced microbial contamination; with reductions of 0.75–0.79 log CFU/g for *L. monocytogenes* (compared to 0.17 log CFU/g in control) and about 0.59–0.64 log CFU/g for total viable count.	A strong deterioration in the sensory quality of the product was observed; the colour and odour of salmon were significantly affected after treatments, whereas the texture and firmness of tissue were not significantly changed.	[82]
Catfish fillets	Near-neutral electrolysed water (anolyte) (pH 6.0 to 6.5 ± 0.02; oxidation reduction potential > 700 mV; residual chlorine concentration of 10 to 200 ppm) was applied for 3 min.	*Salmonella* spp. *L. monocytogenes*	Treatment with anolyte resulted in a 1 log reduction for Salmonella and this reduction was maintained even after 8 days of refrigerated storage.	No reduction in *L. monocytogenes* was observed.	[119]
Shrimps	AEW was prepared by electrolysis of 0.1% sodium chloride (NaCl) solution and frozen.	Autochthonous microbiota	AEW ice showed a good ability to limit pH and colour changes and the formation of total volatile basic nitrogen (TVBN). Bacterial growth was controlled (reduction >1.0 log CFU/g, i.e., >90%) after 6 days.	No negative effects were observed.	[120]
Farmed puffer fish (*Takifugu obscurus*)	The WAEW was generated by electrolysing a solution of hydrochloric acid (3%). The hypochlorous acid content, oxidation–reduction potential and pH value were 21 ppm, 947.6 mV and 6.1, respectively. The treatment was combined with modified atmosphere and vacuum packaging.	Total viable count H_2_S-producing bacteria (including *Shewanella putrefaciens*) *Pseudomonas* spp. Lactic acid bacteria	The combined effect of WAEW and MAP of 60% CO₂/5% O₂/35% N₂ proved to be the most effective in maintaining better quality and prolonging shelf-life to 18 days against 9 days of untreated samples.	No negative effects were found.	[121]
Brown sole (*Pleuronectes herzensteini*)	Slightly acidic electrolysed water was produced by electrolysis of a 6% HCl solution and used in combination with a 5% *w/v* grapefruit seed extract solution after freezing.	*Pseudomonas* spp. Total viable count H_2_S-producing bacteria	Microbial growth was controlled, shelf-life was extended to 12–13 days and sensory characteristics were improved.	No negative effects were found.	[102]

## 4. Some Critical Considerations

Nowadays, consumers continue to demand healthy foods which must be safe while preserving their naturalness. To meet this trend, the food industry could find a precious ally in non-thermal food processing technologies, especially those discussed in this review. In fact, against the negative effects associated with thermal food processing methods, i.e., the high operational costs and alteration of food nutrient components, innovative non-thermal food processing methods are able to cause microbiological inactivation without or with little use of heat.

In general, all the innovative methods analysed in this review have shown a good effectiveness in the control microbial growth in fish products, maintaining their organoleptic, nutritional and sensory characteristics. In addition, these technologies have the great advantage to have a lower energy consumption and shorter production times. On the other hand, their application has already some limitations, as not all these methods are in the same development stage. Each method, as also shown in Figure 1, has some benefits and some issues to address; moreover, some of them exerted a mild effect, while, for others, the extent of antimicrobial effect is higher.

In particular, as mentioned above, a strong limitation of NTAP is lipid oxidation, but some recent studies have demonstrated that this negative aspect could be minimised by using antioxidant compounds in combination [24,25,26,34,35].

US and EW, used individually, do not particularly influence the sensory parameters of fish products, but do not lead to a significant decrease in microbial load. Additionally, in this case, a hurdle approach suggests that the antimicrobial action could be enhanced by combining the cavitation effect of US and the chlorine effect of EW [122]. All the analysed techniques could also be combined with modified atmosphere (MAP), or the addition of antimicrobial compounds on functional edible films, which would improve the sensory parameters and microbiological quality of the fish products [122].

Data reported in the literature point out that the decontamination effect of PEF is low; moreover, this technique appears not suitable for fish products, due to the structural changes it may produce [67]. In contrast, using PEF to extract bioactive compounds from fish waste is promising [57].

An overall analysis allows us to define the following key-points:The use of NTAP in fish products is limited by the synthesis of off-flavours and off-odours compounds (due to lipid oxidation) that affect the product quality;PEF, similar to US, applied to pre-packed and liquid fish products, has shown a great potential to produce high quality products maintaining optimal appearance and sensory attributes. However, it has been little explored and further studies should be performed to consider a suitable application in fish industries;PL is able to inactivate important pathogens, such as *L. monocytogenes*, *E. coli* and *Salmonella* spp., but it greatly compromises the sensory quality;EW studies have shown minor quality degradation compared with other treatments, highlighting EW as the most promising technology;The synergistic effects among non-thermal and other technologies showed great potentials in the fish industry, enhancing the product quality throughout the application of hurdle technology;From an economic point of view, US technology should be extremely beneficial for fish processors.

## 5. Conclusions

Non-thermal technologies developed in recent decades have received much attention in the food industry, showing great potential compared to traditional preservation methods, although some limitations with respect to sensory attributes have been highlighted when used under extreme working conditions.

These limitations could be overcome by using the technologies in a synergistic and combined way, thus increasing microbiological safety and sensory quality. However, further studies are needed to implement these technologies on an industrial level, as many of them are still used on a laboratory scale.

Future investigations should be aimed to develop one (or more) hurdle-like non-thermal method combining different processing methods in order to help the food industries to choose the best technology that meets their food production requirements and the modern consumers’ demand.

## Figures and Tables

**Figure 1 foods-10-02854-f001:**
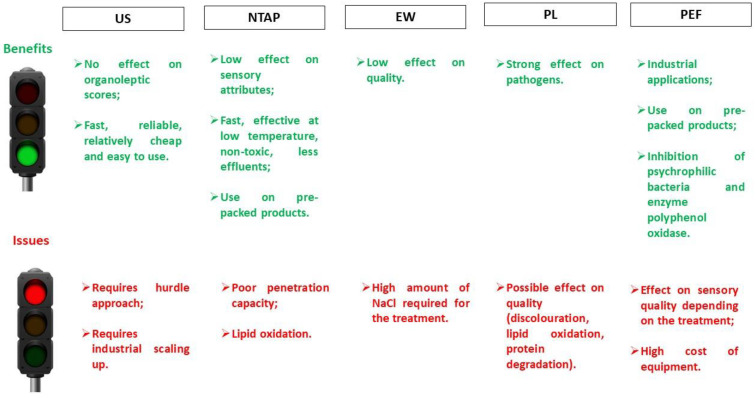
Benefits and issues of non-thermal approaches to prolong the shelf-life of fish. US, ultrasound; NTAP, non-thermal atmospheric plasma; EW, electrolyzed water; PL, pulsed light; PEF, pulsed electric field.

**Table 1 foods-10-02854-t001:** Types of spoilage, causes and main changes observed in spoiled fish.

Types of Fish Spoilage	Causes	Changes
Biological		
➢ Enzymatic	Glycolytic enzymes Autolytic enzymes Cathepsins Chymotrypsin, trypsin, carboxy-peptidases Calpain Collagenases Trimethylamine Oxide (TMAO) demethylase	Lactic acid production, flavour changes in fish flesh (nucleotide degradation), belly-bursting, colour change (black discoloration, yellowing of fish flesh, brown discoloration)
➢ Microbial	Specific Spoilage Organisms (SSO) (*Pseudomonas, Shewanella, Photobacterium, Acinetobacter, Aeromonas, Moraxella,* H_2_S producing bacteria) Pathogenic bacteria: - Indigenous bacteria (*Clostridium, Vibrio* sp., etc.) - Non-indigenous bacteria (*Salmonella* sp., *Escherichia coli*, Shigella)	Loss of juiciness, firm texture, discolouration, and formation of ammonia-like off-flavours due to TMA production
Chemical	Oxidative rancidity	Rancid flavour and odour, texture changes
	Non- enzymatic oxidation	Discolouration

## Data Availability

Not Applicable.

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
