# Peer review of "Innovative Preservation Methods Improving the Quality and Safety of Fish Products: Beneficial Effects and Limits"

_foods, 2021, doi:10.3390/foods10112854_

Round 1

Reviewer 1 Report

In general, an interesting review of fish preservation, but without too much novelty not already found in more generic reviews of novel preservation technologies. The authors use a lot of text to come to the point (the innovative technologies). Seven pages on generic information about seafood, how they spoil and traditional preservation.  The authors seem to assume there is equality between reduced microbial activity and improved quality, that is not necessarily correct. A product can be spoiled or made unacceptable in many other ways than only microbial activity even if often for seafood this is the limiting factor. In the tables you focus on the microorgansims tested, but it is much more interesting to see the effect measured by sensory analyses.  A lot of the results in the tables are also mentioned in the text, so maybe many of the tables should be reduced considerably or even deleted as you summarize the findings quite nicely in Figure 1.

Specific comments of the innovative technologies (the first 7 pages could be shortened to one page introduction and referring to previous written reviews on the topics of fish, fish spoilage, and traditional preservation technologies):

3.2.1 NTAP

The authors fail to mention this method is not currently allowed to be used on foods. A highly relevant info to include. Also, you should emphasise the increase oxidative reactions, and how they negatively influence quality. All the results in table shows a quite limited positive effect of NTAP, in the range of 1-2 log units (don’t report inactivation as percentage, 90% is still only 1 log reduction, and 99% = 2 log). And when you write “shelf-life was increase to xx days” you have to write from what shelf life also. “A 5 minute treatment extended shelf life to 12 days..from 11? Days for the control). And for those reports who does not report any negative effects (or limits as you call it), have they not measured sensory or oxidative changes, and if so, how do they determine shelf-life.

High installation cost for NTAP is mentioned as a drawback, in my experience this is not true, this is one of the cheapest investments in processing technologies we have done, and much cheaper than the other technologies mentioned, e.g HPP, PEF, thermal processing retort.

3.2.2 HPP

HPP in general inactivates microorganisms with increasing pressure, to have any significant effect on the shelf-life usually pressures above 300 MPa is needed, and more typical around 500-600 MPa (and even higher is you want to have any spore-control). This high pressure will denaturate proteins and give a hardened texture and a cooked appearance in a fresh or untreated seafood products. When you report no such changes, it is because you are looking at products than have gone through a protein denaturation either by cooking (like shrimps) or salting (desalted cod fillets) og HPP will not re-denaturate already denaturated proteins. And you mention smoked rainbow trout not being altered by HPP, this is true if we test hot-smoked trout, but not true if cold-smoked (internal colour change til happened)

HPP still is a costly process both to invest in and use, so mainly targeting the premium products.

3.2.3 PEF

PEF is not a good preservation method for solid food, due to the high energy required to inactive microorganisms, and as a preservation method currently only applicable for liquids. However, PEF, has a great potential to increase mass-transfer in extraction, texturizing, drying/rehydrating, salting/desalting, thawing and freezing, and most PEF publications within seafood would be found in these fields. But this is not the topic for the review, and maybe with only one paper about PEF preservation of seafood this section should be shortened considerably.

3.2.4 Pulsed light

Will this method really work on a seafood product with a non-even surface. How do you handle the shadow-effects? I can see the point of treating uniform surfaces such as packaging materials, but not food. Most of the effects of pulsed light is reported in control lab studies, and not applicable in RL.

3.2.5 Ultrasound

Again, not a good preservation method for seafood (or any product). Mostly interesting for increased mass-transfer as mentioned for PEF.

3.2.6 EW

Not familiar with this technology. Sound like something that would need Novel food acceptance to be used in Europe, so maybe a legislation discussion here (or about the innovative technologies in general), quite a few of them will be affected by the novel foods regulations in EU.

Conclusions

As summarized in your Figure, most of these technologies gives at best a 1-2 log unit reduction in bacterial load (except HPP) – with the potential of increase shelf-life 1-2 days maybe, if not oxidative or sensory changes in fact shortens the shelf-life as the products are unacceptable but still of better microbiological quality.

Strange you have not but some more effort to discuss and review into the best way to increase shelf-life of seafood, that is temperature control. Lowering the temperature with one degree during the whole shelf-life would give a better effect than most of the technologies mentioned, and e.g going from 4 degrees C to 0 degrees for a temperate fresh fish would double the shelf-life. No of the novel technologies can do this. And then you can even go to sub-zero temperature before the initial freezing points, for even better effects. Also not covered by this review, is the area of active packaging, novel modified atmosphere packaging (other gases, gas emitters, oxygen aborbers, souble gas stabilisation a.o)

Overall, the review is well readable in a quite good English language, but not too novel.

Reviewer 2 Report

The manuscript provides a long work including a lot of information. The way it has been presented includes too many items so that all of them are treated in a relatively divulgate way. I have many doubts about this manuscript, extended modifications ought to be carried out.

Abstract

Some general aspects could be avoided from this section. Contrary, new methods to be developed ought to be somewhat presented in a more extended way. Also, some explanation on the reason for choosing such methods could be included.

Introduction

Line 17: Replace unsaturated by ω3-PUFA.

Line 45: Replace has by have.

Line 140: The browning development in crustacean is not included here.

I think the part of composition description ought to be widely shortened.

Section 2

Sub-section 2.2.1.: The title ought to also include biogenic amines, not only TMA.

2.3. sub-section:

Phospholipids also lead to FFA formation.

Salting is explained relatively in detail while other processing are mentioned very scarcely. My personal opinion is that all these explanations of traditional methods ought to be eliminated.

HHP Sub-section:

Presentation of this technology is not satisfactory. This technology has been employed for several decades in aquatic food. It is not new. No comment is provided concerning its possible negative effects on lipid oxidation and the resulting non-enzymatic browning.

Line 549-550: This is not correct. Sensory problems have been largely mentioned if pressure conditions are not optimised.

Line 869: HHP cannot be considered a promising technology. It has been used from nineties from last century.

Examples to be mentioned would imply a much longer list of manuscripts. In the current case, examples are mentioned without any order. I mean fresh, frozen, salted smoked, etc. products are mixed without making differences on the results according to the kind of final aquatic food.

I think the part on HHP ought to be eliminated, so that the authors could concentrate on the advanced and promising preserving technologies. If the authors insist on including HHP, much more information ought to be included. The authors should then review recent revisions published on HHP on seafood.

Reviewer 3 Report

The manuscript aims to provide a comprehensive review of the current innovations in the field of fish processing.

Section 1 should be rewritten and enriched, since currently it is based mainly on references [1] and [2], apart from the last paragraph. Please give a more comprehensive approach of the state of the art.

Line 13: Replace with “fishery” with “fish”

Line 45: Do you mean “market value”?

Section 2 is a brief description of the common spoilage mechanisms of fish. This is based on conventional approaches, already established during the last decades; No critical evaluation of the assumptions reported is provided.

The same comment is valid for Section 3. Overall, thew first 373 lines of the manuscript are presented well established information without critical evaluation, which is more common to a book chapter that a high quality and critical review article.

The innovative technologies in Section 3 is provided in a more explicit way, since it presents also the limitations of the presented methods in the respective Tables. However, the discussion in the main body of the manuscript is poor and the authors should elaborate more in this section. Although the authors present the existing literature in tables in a very analytical and explanatory manner, which is very useful, at the same time they should not use the same references throughout the manuscript. This repetition is tiring.  

The mode of action of the discussed preservation methods should have been explained more detailed.

Additionally, the information provided in Figure 1 are not accurate in several cases. For example, specific and high cost equipment is also appropriate for HHP, which is not stated in the Figure. Use on pre-packed products is also possible for cold plasma applications, which is not reported here. Industrial applications have not been reported for cold plasma on fish products etc.   

Reviewer 4 Report

The recommendations are made in order to improve the quality of the text in the document.

Line 16: .. .PP, A, and D)…

Line 17: …selenium, and sodium)…

Line 17: unsaturated fats (mainly ……).

Line 44: …3,000…

Line 60: delete space between the number value and %.

Line 91: …Blue…

Line 100: …White…

Line 128: …post-mortem…note: change trough the manuscript

Line 133: non–protein by non-protein

Line 133: …pH > 6.0..

Line 177: …, and Vibrio)…

Line 178: …, and Lactobacillus)..

Line 179: ….Alteromonas, and..

Line 182: …, and Yersinia.

Line 215: post-mortem

Line 219: 'fishy smell'

Line 221: decarboxylation, including…., and spermine

Line 244-249; 290-296; 297-299; 300-305; 306-317; 348-358; 391-396; 397-405; 422-426; 427-434; 469-475; 476-478; 479-483; 492-503; 565-587; 715-717; 741-745: it were missed the references of the paragraphs

Line 377: ..smell, and..

Line 380: shelf-life

Line 438: microorganisms

Line 453: delete space between the number value and °C

Table 1: reference 49….m2…..

Table 1: reference 41,42,47….2°C…..minute or min? sometimes appears in the abbreviate form

Table 1: reference 48….25°C…

Line 533: …50,70, and 100…

line 545: shelf-life

Line 597: …, and shape

Line 599: (HILP), and

Line 612: , and

Table 2: reference 56…5.0, and 1.5

Table 2: reference 74…25°C

Table 2: reference 75…min or minutes?..shelf-life

Table 2: reference 62…min or minutes?.. > 6 log10

Table 2: reference 61…2°C…min or minutes

Table 2: reference 61,60…idem

Table 2: reference 59…0.5%, v/v….80°C

Table 2: reference 57…50, 70, and 100%

Line 636: processing, and

Line 647: Gram

Line 676: O2

Line 685: filtration, and

Line 688: physical, and

Line 197: plates, and

Line 767: S. aureus

Table 3: reference 60….50, 60 and 70%

Table 3: reference 93….minutes or min

Line 789: C. jejuni

Line 791: log

Table 4: 50 and 15% (v/v)

Table 104: sometimes bacteria's appears with or without the abbreviate form; minutes or min

In the references, some article titles appear with capital letters for each word, the year of each reference is not marked in bold (check the authors' guide)

Round 2

Reviewer 2 Report

I think the manuscript has been reduced and performed substantially. Its content seems now to be more related to the real objective of the manuscript, i.e., to show the advantages of applying advanced non-thermal technologies on fish products. However, I think several performances ought to be done.

Abstract

This section is now very short and not providing the interest in the advanced technologies. Preliminary information related to such advanced technologies chosen ought to be provided, as well as their possible advantages.

Introduction

It is not necessary to provide so much information about composition of fish products.

Fish spoilage in fish products is known to be important. However, its description in this section is too long. Just the basic aspects ought to be mentioned.  

Furthermore, description of traditional methods is out of scope. In my opinion, this part ought not to be placed here.

Reviewer 3 Report

The authors addressed the reviewer's suggestion, however the manuscript is still far from being appropriate for publication as a review article. No critical evaluation is provided regarding the alternative technologies that are presented relevant to fish and fish products. This is the main aspect of a scientific review article and the authors should elaborate more on the discussion rather than the single presentation of the different methods.
